# Persistence of sleep difficulties for over 16 years amongst 66,948 working-aged adults

**Mikhail Saltychev**[1]*, **Juhani Juhola**[1], **Jari Arokoski**[2], **Jenni Ervasti**[3], **Mika Kivimäki**[3,4,5], **Jaana Pentti**[6], **Sari Stenholm**[6,7], **Saana Myllyntausta**[6], **Jussi Vahtera**[6]

**1** Department of Physical and Rehabilitation Medicine, Turku University Hospital and University of Turku, Turku, Finland, **2** Department of Physical and Rehabilitation Medicine, Helsinki University Hospital and University of Helsinki, Helsinki, Finland, **3** Finnish Institute of Occupational Health, Helsinki, Finland, **4** Clinicum, Faculty of Medicine, University of Helsinki, Helsinki, Finland, **5** Department of Epidemiology and Public Health, University College London Medical School, London, United Kingdom, **6** Department of Public Health, University of Turku and Turku University Hospital, Turku, Finland, **7** Centre for Population Health Research, University of Turku and Turku University Hospital; Turku, Finland

* mikhail.saltychev@gmail.com

**Data Availability Statement:** All relevant data are within the paper and its Supporting Information files.

## Abstract

The objective was to investigate the persistence of sleep difficulties for over 16 years amongst a population of working age. In this prospective cohort study, a group-based trajectory analysis of repeated surveys amongst 66,948 employees in public sector (mean age 44.7 [SD 9.4] years, 80% women) was employed. The main outcome measure was sleep difficulties based on Jenkins Sleep Scale (JSS). Up to 70% of the respondents did not experience sleep difficulties whereas up to 4% reported high frequency of notable sleep difficulties through the entire 16-year follow-up. Heavy drinking predicted sleep difficulties (OR 2.3 95% CI 1.6 to 3.3) except for the respondents younger than 40 years. Smoking was associated with sleep difficulties amongst women younger than 40 years (OR 1.2, 95% CI 1.0 to 1.5). Obesity was associated with sleep difficulties amongst men (OR 1.9, 95% CI 1.4 to 2.7) and women (OR 1.2, 95% CI 1.1 to 1.3) of middle age and amongst women older than 50 (OR 1.5, 95% CI 1.2 to 1.8) years. Physical inactivity predicted sleep difficulties amongst older men (OR 1.3, 95% CI 1.1 to 1.6). In this working-age population, sleep difficulties showed a great persistence over time. In most of the groups, the level of sleep difficulties during the follow-up was almost solely dependent on the level of initial severity. Depending on sex and age, increasing sleep problems were sometimes associated with high alcohol consumption, smoking, obesity and physical inactivity, but the strength of these associations varied.

## Introduction

The concept of "sleep difficulties" may include mild dissatisfaction in sleep duration or quality, as well as severe insomnia and other clinically significant sleep disorders. The prevalence of sleep difficulties varies from a few percent up to 60% amongst adults, depending on the study population and diagnostic criteria for sleep difficulties [1–8]. However, it is widely agreed that

**Funding:** This study was supported by funding granted by the Academy of Finland (Grants 286294, 294154, 319246 and 332030 to SS; 633666 to MK; 321409 and 329240 to JV); the Finnish Ministry of Education and Culture (to SS); NordForsk (to MK and JV); the UK MRC (Grant K013351 to MK); The Finnish Environment Fund (Grants 118060 to SS); Hospital District of Southwest Finland (to SS).

**Competing interests:** The authors have declared that no competing interests exist.

sleep difficulties are a significant problem associated with higher comorbidity and mortality and is the cause of substantial economic loss including costs of work disability [9, 10].

Sleep difficulties have been found to be associated with female gender, physical inactivity, excessive alcohol consumption and insufficient amount of sleep hours [3, 6, 11–18]. The persistence of sleep difficulties has previously been studied mostly amongst children and elderly [19–22]. Only a few studies, cross-sectional or with short-term follow-ups, focused on general or working-age population [2]. A study on a cohort of people of working-age stated that the prevalence of occasional insomnia-related symptoms could be around 40% to 45% and, on a population level, that estimate prevails over time [23]. A recent study among a general population has reported that insomnia might be a very persistent condition [24]. However, several important questions remain. For example, how persistent are sleep problems in a general population? Does a person usually experience them for a limited period of time, or are they likely to experience the same amount of sleep difficulties more or less permanently? Does a baseline severity of sleep difficulties pertain also in the future? What factors may affect the trajectories of sleep complaints over time?

To address these questions, the objectives of this study were to investigate a) if sleep difficulties persist or whether the severity of these difficulties are likely to change over time, b) if different subgroups with different trajectories of sleep difficulties could be defined; and c) what modifiable factors might be associated with the severity of sleep difficulties.

## Methods

Participants were drawn from the Finnish Public Sector (FPS) cohort study of employees of 10 towns and 6 hospital districts. Data included responses to five questionnaire surveys administered to the FPS sub-cohorts in 4-year intervals from 2000 to 2017 (average response rate 70%). For this study, the baseline was the response given in 2000 or in 2004. All the respondents have approved a written informed consent. The study did not include minors. The ethics committee of the Hospital District of Helsinki and Uusimaa approved the study plan and the informed consent form (registration number HUS/1210/2016).

S1 File contains a complete description of the dataset (in Finnish). The data are not publicly available due to legislative restrictions, as the data contains information that could compromise the privacy of the research participants. The restrictions upon the dataset were imposed by the data owner, Finnish Institute of Occupational Health. The deidentified data that support the findings of this study are available on reasonable request from the corresponding author MS or directly from the data owner, Finnish Institute of Occupational Health–principal investigator JE Jenni.Ervasti@ttl.fi.

Age, gender, body mass index (BMI), level of physical activity, alcohol consumption, and smoking status were measured at the time of the first response. Age was defined in full years. The BMI was defined as weight/height$^2$ and dichotomized to indicate obesity if BMI $\geq$30 kg/m$^2$. The level of physical activity was calculated from the survey responses, converted into metabolic equivalent of task (MET) and dichotomized based on the cohort's quartiles as "low physical activity"–the lower quartile vs. others. Alcohol consumption was obtained from the survey and converted into g/week, and >210 g of pure alcohol per week was considered a cut-off for excess alcohol consumption (no/yes). Smoking status was dichotomized as current smoking yes/no.

The JSS is a four-item questionnaire to follow common sleep problems in clinical areas. Four items evaluated, in the last month, the difficulty to fall asleep, wake up at night, difficulty to stay asleep and wake up exhausted in the morning. Each item is rated on a Likert-like scale from zero to five, where zero is "never", 1 is "1–3 days", 2 is "4–7" days, 3 is "8–14 days", 4 is

"15–21 days" and 5 is "22–28 days". The total score is a simple sum of all four items' scores zero (no sleep problems) to 20 (most sleep problems). In order to include also incomplete responses to the JSS, in this study, a total score was substituted by the average score of answered items (0.0 to 5.0) using a method of person mean imputation [25]. The JSS has been one of the most commonly used questionnaires in epidemiological sleep studies 1–4. It has been found to be valid and reliable amongst patients with different health problems as well as in large non-clinical populations [26–29].

## Statistical analysis

The estimates were reported as means and standard deviations or as absolute numbers and percentage when appropriate. Group-based trajectory modeling was used to investigate the developmental trajectory (a course of outcome over time) of the severity of sleep disorders measured by the JSS. This method is a form of finite mixture modeling for analyzing longitudinal repeated measures data [30–32]. While conventional statistics show a trajectory of average change of outcome over time, group-based trajectory modeling is able to distinguish and describe subpopulations (clusters) existing within a studied population. The trajectories of such subpopulations may differ substantially from each other and from the average trajectory of the entire population. In this study, the procedure consisted of the following steps:

1. Censored (known also as 'regular') normal modeling was used with minimum and maximum values set at the lowest and the highest possible JSS scores (0 to 5).

2. The studied population was divided into 6 gender-age groups: <40, 40–49, and 50+ for men and women.

3. The number of groups may be defined by the size of the data set measuring in two dimensions: the number of cases and the number of repeated measures. There are no common recommendations on the number of trajectory groups. In theory, it can be any number from one up to the number of cases. Previous research has suggested breaking a sample down below 300 cases may not add significant information. In this study, we pre-agreed that the smallest group should be around 3% of the entire sample or the size of the smallest group should be around 300 cases [33]. We also pre-agreed that the number of trajectory groups will be the same for all gender-age groups to ease the interpretation of the results. This way, six cluster groups were identified for each gender-age group. The goodness of model fit was judged by running the procedure several times with a number of subpopulations starting from one up to six. A cubic regression was applied.

4. The Bayesian Information Criterion (BIC), Akaike information criterion (AIC) and average posterior probability (APP) were used as criteria to confirm the goodness of fit.

Odds ratios (ORs) were used to describe the associations of risk factors and the probability of being classified into a particular cluster. The ORs were accompanied by their 95% confidence intervals (95% CIs). The analyses were performed using Stata/IC Statistical Software: Release 16. College Station (StataCorp LP, TX, USA). The additional Stata module 'traj' was required to conduct group-based trajectory analysis. The module is freely available for both SAS® and Stata software (Jones and Nagin 1999; 2013).

## Results

Of the 66,948 respondents, 53,541 (80%) were women and 13,407 (20%) were men. The average age was 44.7 (SD 9.4) years. Excessive alcohol consumption was reported by 36%, smoking by 12%, low physical activity by 24%, and obesity by 42% of the respondents. Table 1 illustrates

**Table 1. The goodness of fit of group-based trajectory analysis models.**

| Model | Smallest group size | BIC[1] | AIC[2] | APP[3] |
|---|---|---|---|---|
| Men <40 (n = 4,286) | | | | |
| 1-cluster | 100% | 20,710 | 20,694 | 1.0 |
| 2-cluster | 21% | 19,224 | 19,192 | 0.87 to 0.94 |
| 3-cluster | 10% | 18,764 | 18,716 | 0.80 to 0.86 |
| 4-cluster | 4% | 18,632 | 18,569 | 0.76 to 0.82 |
| 5-cluster | 4% | 18,527 | 18,447 | 0.65 to 0.84 |
| **6-cluster** | **3%** | **18,469** | **18,374** | **0.65 to 0.78** |
| Men 40–49 (n = 4,274) | | | | |
| 1-cluster | 100% | 24,056 | 24,040 | 1.0 |
| 2-cluster | 28% | 22,008 | 21,976 | 0.90 to 0.95 |
| 3-cluster | 14% | 21,390 | 21,342 | 0.84 to 0.89 |
| 4-cluster | 9% | 21,190 | 21,126 | 0.79 to 0.88 |
| 5-cluster | 6% | 21,082 | 21,003 | 0.71 to 0.82 |
| **6-cluster** | **3%** | **20,997** | **20,902** | **0.71 to 0.78** |
| Men > 49 (n = 4,847) | | | | |
| 1-cluster | 100% | 28,438 | 28,423 | 1.0 |
| 2-cluster | 34% | 26,191 | 26,159 | 0.90 to 0.94 |
| 3-cluster | 13% | 25,520 | 25,471 | 0.85 to 0.89 |
| 4-cluster | 8% | 25,322 | 25,257 | 0.80 to 0.86 |
| 5-cluster | 3% | 25,279 | 25,198 | 0.75 to 0.84 |
| **6-cluster** | **3%** | **25,149** | **25,052** | **0.73 to 0.80** |
| Women <40 (n = 17,751) | | | | |
| 1-cluster | 100% | 93,373 | 93,353 | 1.0 |
| 2-cluster | 26% | 86,571 | 86,532 | 0.88 to 0.94 |
| 3-cluster | 8% | 85,010 | 84,951 | 0.80 to 0.87 |
| 4-cluster | 3% | 84,511 | 84,433 | 0.73 to 0.85 |
| 5-cluster | 4% | 84,099 | 84,002 | 0.65 to 0.83 |
| **6-cluster** | **2%** | **83,899** | **83,783** | **0.63 to 0.76** |
| Women 40–49 (n = 18,010) | | | | |
| 1-cluster | 100% | 113,267 | 113,248 | 1.0 |
| 2-cluster | 32% | 104,238 | 104,199 | 0.90 to 0.94 |
| 3-cluster | 12% | 101,825 | 101,766 | 0.84 to 0.88 |
| 4-cluster | 6% | 101,157 | 101,079 | 0.77 to 0.87 |
| 5-cluster | 7% | 100,687 | 100,589 | 0.67 to 0.84 |
| **6-cluster** | **3%** | **100,344** | **100,227** | **0.69 to 0.76** |
| Women > 49 (n = 17,780) | | | | |
| 1-cluster | 100% | 114,688 | 114,668 | 1.0 |
| 2-cluster | 33% | 105,830 | 105,791 | 0.90 to 0.94 |
| 3-cluster | 13% | 103,340 | 103,281 | 0.84 to 0.88 |
| 4-cluster | 6% | 102,599 | 102,521 | 0.80 to 0.85 |
| 5-cluster | 6% | 102,091 | 101,994 | 0.70 to 0.83 |
| **6-cluster** | **3%** | **101,782** | **101,666** | **0.70 to 0.83** |

The chosen models are shown in bold.

[1] BIC = Bayesian Information Criterion

[2] AIC = Akaike information criterion

[3] APP = average posterior probability.

the path of defining the final set of six trajectory groups. The table shows that the goodness of fit increased with every step from one- to six-group model. Six-cluster models with cubic regression demonstrated good fit for each gender-age group.

For each gender-age group, the identified trajectories followed a similar pattern (Fig 1). *95% confidence limits are shown as dot-lines."* Two trajectories with least sleep problems at baseline without any substantial change during the follow-up accounted for 50% to 70% of the respondents. For further analysis, these two trajectories were combined into one cluster, which served as a reference cluster. Also, for each gender-age group, there was a cluster/trajectory with consistently high frequency of sleep difficulties (every night), which represented 2% to 4% of a particular group. Almost every group demonstrated two or three trajectories with either decreasing or increasing frequency of sleep difficulties.

As shown in Table 2, many risk factors were inconsistently associated with both sleep difficulties and good sleep. However, there were several statistically significant unequivocal associations. Except for men and women younger than 40 years, heavy drinking predicted either steadily high or increasing sleep difficulties–OR varied from 1.15 (95% CI 1.01 to 1.31) up to 2.30 (95% CI 1.62 to 3.27). Smoking was associated with worsening sleep difficulties amongst women younger than 40 years–(OR 1.23, 95% CI 1.04 to 1.46). Low physical activity predicted sleep difficulties amongst men older than 50 years (OR 1.29, 95% CI 1.07 to 1.56). Obesity was associated with sleep difficulties amongst men (OR 1.92, 95% CI 1.36 to 2.70) and women (OR up to 1.19, 95% CI 1.08 to 1.30) of middle age and amongst women older than 50 years (OR 1.47, 95% CI 1.22 to 1.77).

## Discussion

This prospective survey-based cohort study investigated the persistence of sleep difficulties experienced by 67,000 employed people depending on age and gender during a 16-year follow-up. Additionally, the study evaluated the associations of four modifiable risks with different trajectories of changes in sleep, using the JSS. From 50% to 70% of the respondents in each group had no, or only mild sleep difficulties through the entire follow-up. There were several different trajectories (responsible for a small part of the studied cohort) showing increasing or decreasing frequency of sleep difficulties. In most of the groups, the severity of sleep difficulties remained unchanged during the follow-up and this severity was depending only on the initial level. Except for the respondents younger than 40 years, heavy drinking predicted either steadily worse or worsening sleep difficulties. Smoking was associated with worsening sleep difficulties amongst young women. Physical inactivity was associated with sleep difficulties amongst older men. Obesity was associated with sleep difficulties amongst middle-age respondents and amongst older women.

It is noteworthy that trajectory analysis provides only an approximation of changes in sleep patterns. The method shows probable trends in a particular population. In this study, the population was limited to employed people of working age–mostly between 40 and 50 years. The majority of the participants were women, which reflects the sex distribution in public sector employees in Finland. While this may limit the generalizability of our findings, the size of the studied cohort and the longitudinal design strengthen validity of the observed associations. Also, the study offered a sophisticated instrument–group-based trajectory analysis–to evaluate fluctuations within a cohort's clusters, which with more basic approaches often remain undetected.

The observed associations between sleep difficulties and physical inactivity and excessive alcohol consumption were in line with previous reports [3, 6, 11–18]. Previous studies are comparable with the present study with reservations as they have usually approached the

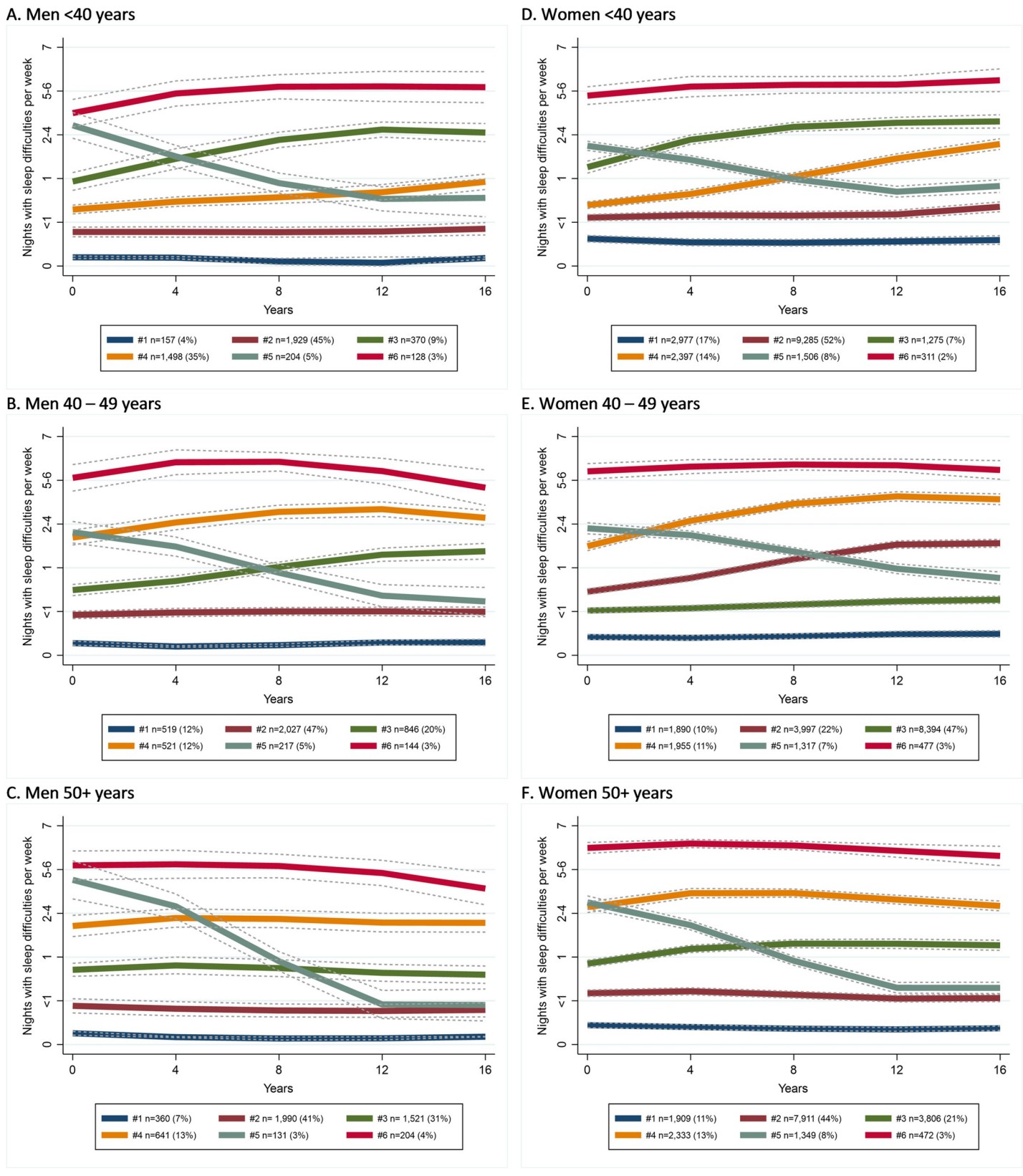

**Fig 1. Trajectories of the JSS by gender-age groups.**

**Table 2. The strength of prediction amongst modifiable risks and probability of being placed into a particular cluster.**

| Trajectories | Risk factors | | | | | | | | | | |
|---|---|---|---|---|---|---|---|---|---|---|---|
| | Heavy drinking | | | Smoking | | | Low physical inactivity | | | Obesity | | |
| | OR | 95% CI | | OR | 95% CI | | OR | 95% CI | | OR | 95% CI | |
| Men <40 years (Fig 1A) | | | | | | | | | | | | |
| Steadily average sleepers (1 n/w [a]) | **1.27** | **1.11** | **1.45** | 1.02 | 0.85 | 1.23 | 1.16 | 0.97 | 1.40 | **1.19** | **1.04** | **1.35** |
| Steadily worst sleepers (5–6 n/w) | **1.98** | **1.37** | **2.85** | 1.51 | 0.96 | 2.36 | **1.82** | **1.19** | **2.79** | **1.43** | **1.00** | **2.05** |
| Worsening sleepers (1 → 2–4 n/w) | 1.06 | 0.85 | 1.32 | 1.24 | 0.92 | 1.66 | **1.54** | **1.16** | **2.04** | **0.78** | **0.62** | **0.98** |
| Improving sleepers (2–4 → 1 n/w) | **1.49** | **1.11** | **1.98** | 1.10 | 0.74 | 1.63 | **1.93** | **1.37** | **2.72** | 1.07 | 0.80 | 1.42 |
| Men 40–49 years (Fig 1B) | | | | | | | | | | | | |
| Steadily average sleepers (2–4 n/w) | **1.62** | **1.34** | **1.96** | 0.93 | 0.72 | 1.19 | 1.21 | 0.97 | 1.50 | 1.19 | 0.99 | 1.44 |
| Steadily worst sleepers (5–6 n/w) | **2.30** | **1.62** | **3.27** | 1.06 | 0.69 | 1.64 | 1.15 | 0.78 | 1.70 | **1.92** | **1.36** | **2.70** |
| Worsening sleepers (<1 → >1 n/w) | 1.03 | 0.89 | 1.21 | 1.03 | 0.84 | 1.26 | 1.14 | 0.95 | 1.37 | 1.02 | 0.87 | 1.19 |
| Improving sleepers (2–4 → 1 n/w) | 0.93 | 0.71 | 1.23 | 0.83 | 0.56 | 1.23 | 1.32 | 0.97 | 1.80 | 0.90 | 0.68 | 1.19 |
| Men 50+ years (Fig 1C) | | | | | | | | | | | | |
| Steadily average sleepers (1 n/w) | **1.15** | **1.01** | **1.31** | 0.83 | 0.68 | 1.00 | **1.15** | **1.00** | **1.33** | 0.94 | 0.83 | 1.07 |
| Steadily bad sleepers (2–4 n/w) | 1.12 | 0.94 | 1.33 | 1.04 | 0.81 | 1.33 | **1.29** | **1.07** | **1.56** | 0.93 | 0.78 | 1.11 |
| Steadily worst sleepers (5–6 n/w) | 1.04 | 0.78 | 1.39 | 1.11 | 0.75 | 1.65 | 1.17 | 0.86 | 1.60 | 0.99 | 0.74 | 1.32 |
| Improving sleepers (5–6 → 1 n/w) | 1.01 | 0.71 | 1.44 | 0.95 | 0.57 | 1.59 | **1.45** | **1.00** | **2.10** | 1.20 | 0.84 | 1.71 |
| Women <40 years (Fig 1D) | | | | | | | | | | | | |
| Steadily worst sleepers (5–6 n/w) | 1.10 | 0.88 | 1.39 | 1.05 | 0.74 | 1.49 | **1.51** | **1.17** | **1.96** | **1.31** | **1.05** | **1.64** |
| Worsening sleepers (<1 to 2–4 n/w) | 0.94 | 0.86 | 1.02 | 1.10 | 0.96 | 1.26 | **1.16** | **1.04** | **1.30** | **0.90** | **0.83** | **0.99** |
| Worsening sleepers (1 to 2–4 n/w) | 1.05 | 0.94 | 1.19 | **1.23** | **1.04** | **1.46** | 1.08 | 0.93 | 1.25 | 1.01 | 0.90 | 1.14 |
| Improving sleepers (2–4 → 1 n/w) | **1.25** | **1.12** | **1.39** | 1.03 | 0.87 | 1.22 | **1.17** | **1.02** | **1.33** | **1.19** | **1.07** | **1.32** |
| Women 40–49 years (Fig 1E) | | | | | | | | | | | | |
| Steadily worst sleepers (5–6 n/w) | **1.23** | **1.01** | **1.49** | **2.00** | **1.59** | **2.50** | **1.47** | **1.21** | **1.79** | **1.30** | **1.08** | **1.57** |
| Worsening sleepers (<1 to 1 n/w) | 0.97 | 0.89 | 1.05 | 1.08 | 0.97 | 1.20 | 1.07 | 0.98 | 1.16 | 0.95 | 0.88 | 1.03 |
| Worsening sleepers (1 to 2–4 n/w) | **1.15** | **1.03** | **1.27** | **1.31** | **1.15** | **1.50** | **1.16** | **1.04** | **1.29** | 1.02 | 0.92 | 1.13 |
| Improving sleepers (2–4 → 1 n/w) | 1.02 | 0.90 | 1.16 | **1.29** | **1.10** | **1.52** | **1.21** | **1.07** | **1.38** | 1.03 | 0.92 | 1.16 |
| Women 50+ years (Fig 1F) | | | | | | | | | | | | |
| Steadily average sleepers (1 n/w) | 0.97 | 0.90 | 1.06 | **0.87** | **0.76** | **0.99** | **1.12** | **1.03** | **1.21** | 0.97 | 0.90 | 1.05 |
| Steadily bad sleepers (2–4 n/w) | **1.11** | **1.01** | **1.23** | 0.99 | 0.85 | 1.16 | **1.24** | **1.12** | **1.36** | **1.19** | **1.08** | **1.30** |
| Steadily worst sleepers (5–6 n/w) | **1.29** | **1.06** | **1.57** | 1.28 | 0.96 | 1.71 | **1.27** | **1.04** | **1.54** | **1.47** | **1.22** | **1.77** |
| Improving sleepers (2–4 → <1 n/w) | 0.93 | 0.82 | 1.05 | 1.10 | 0.91 | 1.32 | **1.34** | **1.18** | **1.51** | 1.10 | 0.98 | 1.24 |

Two trajectories with the lowest baseline JSS scores were combined into one cluster ("steadily good sleepers") and used as a reference. Significant results are shown in bold.

[a] Nights per week.

problem from a different point of view. Numerous cross-sectional and longitudinal studies have reported prevalence or risk factors of sleep disorders in different populations [1, 4, 5, 7, 34, 35]. The persistence of sleep difficulties has previously been studied mostly amongst children and adolescents and elderly [19–22]. The respective knowledge regarding general middle-age populations is scarce. A longitudinal study from United Kingdom amongst patients, who had been seen by general practitioners, reported persistence of insomnia in a one-year follow-up [2]. Also, a recent study from Canada reported the persistent nature of insomnia severity in a general population [24]. No longitudinal repeated measure study has focused on the associations between modifiable risks and sleep difficulties in a healthy population.

We observed that sleep difficulties were mostly very persistent. For most of the respondents, the severity of sleep difficulties was depending only on the baseline level of that severity. Even clusters with ascending or descending trajectories showed only mild changes in sleep difficulties. Trajectories with more substantial improvement seen in older persons of both genders might reflect the possible effect of retirement. One might expect much more fluctuation in such a long period of time. It is possible that this phenomenon might be related to the specifics of the study design–the repeated measures described the situation during a particular month at a few rare repeated measures. Hence, short-term variability in sleep difficulties might be undetected.

Further research on large cohorts and with long follow-ups with several repeated measures should be repeated in different populations and settings. Further research should involve different age groups, diverse socio-economic situations, countries and cultures.

## Conclusions

In this working-age population, sleep difficulties showed a great persistence over time. In most of the groups, the level of sleep difficulties during the follow-up was almost solely dependent on the level of initial severity. Depending on sex and age, increasing sleep problems were sometimes associated with high alcohol consumption, smoking, obesity and physical inactivity, but the strength of these associations varied.

## Supporting information

**S1 File.**
(DOCX)

## Author Contributions

**Conceptualization:** Mikhail Saltychev, Juhani Juhola, Jari Arokoski, Jenni Ervasti, Mika Kivimäki, Jaana Pentti, Sari Stenholm, Jussi Vahtera.

**Data curation:** Jenni Ervasti, Mika Kivimäki, Jaana Pentti, Sari Stenholm, Jussi Vahtera.

**Formal analysis:** Mikhail Saltychev, Jenni Ervasti, Mika Kivimäki, Jaana Pentti, Sari Stenholm, Jussi Vahtera.

**Funding acquisition:** Jenni Ervasti, Mika Kivimäki, Sari Stenholm, Jussi Vahtera.

**Investigation:** Mikhail Saltychev, Juhani Juhola, Jenni Ervasti, Mika Kivimäki, Jaana Pentti, Sari Stenholm, Saana Myllyntausta, Jussi Vahtera.

**Methodology:** Mikhail Saltychev, Juhani Juhola, Jari Arokoski, Jenni Ervasti, Mika Kivimäki, Jaana Pentti, Sari Stenholm, Saana Myllyntausta, Jussi Vahtera.

**Project administration:** Jenni Ervasti, Mika Kivimäki, Jaana Pentti, Sari Stenholm, Jussi Vahtera.

**Resources:** Jenni Ervasti, Mika Kivimäki, Jussi Vahtera.

**Software:** Mikhail Saltychev.

**Supervision:** Jenni Ervasti, Mika Kivimäki, Jaana Pentti, Jussi Vahtera.

**Validation:** Mikhail Saltychev, Juhani Juhola, Jari Arokoski, Jenni Ervasti, Mika Kivimäki, Jaana Pentti, Saana Myllyntausta, Jussi Vahtera.

**Visualization:** Mikhail Saltychev, Juhani Juhola, Jari Arokoski, Jenni Ervasti, Mika Kivimäki, Jaana Pentti, Sari Stenholm, Saana Myllyntausta, Jussi Vahtera.

**Writing – original draft:** Mikhail Saltychev.

**Writing – review & editing:** Juhani Juhola, Jari Arokoski, Jenni Ervasti, Mika Kivimäki, Jaana Pentti, Sari Stenholm, Saana Myllyntausta, Jussi Vahtera.

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
