## [Decision Letter · Decision Letter 0]

13 Sep 2021

PONE-D-21-02910Persistence of sleep difficulties over 16 years amongst 66,948 working-aged adultsPLOS ONE

Dear Dr. Saltychev,

Thank you for submitting your manuscript to PLOS ONE. After careful consideration, we feel that it has merit but does not fully meet PLOS ONE’s publication criteria as it currently stands. Therefore, we invite you to submit a revised version of the manuscript that addresses the points raised during the review process (please see below).

We look forward to receiving your revised manuscript.

Kind regards,

Federica Provini

Academic Editor

PLOS ONE

Journal Requirements:

Have the authors made all data underlying the findings in their manuscript fully available?

Reviewer**: No**

Review Comments to the Author

This manuscript examined longitudinal sleep difficulties of adults in Finland. This is the first study to prospectively assess sleep and modifiable health factors in a cohort of adults longitudinally over 2.5 decades. The authors report that sleep difficulties are persistent across the 16 year assessment period. They found specific health factors to be associated with sleep depending upon characteristics of age and sex. A strength of the study is the longitudinal cohort design allowing for repeated measurement and within subjects comparison. The primary weakness of the manuscript are grammatical/spelling/punctuation errors throughout the document that should be addressed for clarity and to improve readability. Additionally, the manuscript would benefit from further description of the manner in which the analyses were conducted with this specific sample in addition to the general description of the analytic techniques that were used. To summarize, this is an interesting research question and a unique population that provides new insights into the relationships between sleep and modifiable health outcomes. Thank you for the opportunity to review this paper.

Abstract and Statement of Significance

1. There are several grammatical errors in the Abstract and the Statement of Significance (e.g. words such as “a” and “the” appear to be missing resulting in some incomplete sentences and impaired readability, the phrase “solely depending” should be changed to “Solely dependent”). These section would benefit from a more thorough review by the authors. The Statement of Significance repeats the last line of the Abstract verbatim and does not explain the significance of this work (e.g. how does this work expand on what is already known about sleep difficulties in adults?).

Introduction

2. There are grammatical errors throughout the Introduction section (as in the Abstract, words such as “a” and “the” appear to be missing; e.g. lines 76-77). Some sentences are awkwardly worded such that readability is impacted (e.g. line 77-78)

Methods

1. There are grammatical errors throughout the Methods section (as in the Abstract, words such as “upon”; e.g. line 122)

2. The Methods section would benefit from a review for errant punctuation and general grammatical/spelling errors (e.g. line 104-10).

3. There is no description of whether informed consent (or a waiver alternative) was obtained from participants.

4. How were physical activity and alcohol consumption dichotomous values determined? If this is based on established norms, citations should be included. Additionally, it might be more accurate to use different cut offs for excessive alcohol consumption based on participant characteristics (e.g. sex or BMI).

5. There is no citation or description or established psychometric properties of the Jenkins Sleep Scale, nor is there a citation for the method for handling missing data.

6. After introducing the JSS abbreviation, Continue to use it through out rather than switching back and forth between the abbreviation and spelling out the full name of the measure in the text.

7. What methods or variables were used to determine the six clusters? What do the clusters consist of? How are they similar or different?

Results

8. It is unclear to me how the cluster models/trajectories (shown in Table 1) relate to the risk factor analyses (shown in Table 2). Further description of the purpose of and relationship between these analyses in either the Methods/Statistical Analysis section or in the Results would be helpful. Discussion

Discussion

9. There are grammatical errors throughout the Discussion section (e.g. lines 166-167, line 174, line 175, line 177, lines 181-183, line 198).

10. Line 178 – The term “instrument” should be replaced with a more accurate term such as “technique,” as the analysis is not physical tool or device as is suggested by the term instrument.

11. Line 201 – change “may” to “should”

Tables and Figures

12. Table 1: the text “The chosen models are shown in bold” belongs in the footnote/caption rather than in the title.

13. Table 2: the text “Two trajectories with lowest baseline JSS scores were combined into one cluster (“steadily good sleepers”) and used as a reference” belongs in the footnote/caption rather than in the title. Additionally, highlighting significant findings in some manner (e.g. bolding text) would be helpful to the reader.

14. The line graph Figures are completely unreadable as the text is blurry.

---

## [Author Response · Author response to Decision Letter 0]

22 Sep 2021

Responses to the comments made by the Editor

Comment 1

and 

Response 1

We have changed the style of figure presentation in the text. Could it be possible to get more information what else should be done?

Comment 2

Please provide additional details regarding participant consent. In the ethics statement in the Methods and online submission information, please ensure that you have specified (1) whether consent was informed and (2) what type you obtained (for instance, written or verbal, and if verbal, how it was documented and witnessed). If your study included minors, state whether you obtained consent from parents or guardians. If the need for consent was waived by the ethics committee, please include this information.

Response 2

We have now added/modified the following text to the Methods section:

“All the respondents have approved a written informed consent. The study did not include minors. The ethics committee of the Hospital District of Helsinki and Uusimaa approved the study plan and the informed consent form (registration number HUS/1210/2016).”

Comment 3

We note that the grant information you provided in the ‘Funding Information’ and ‘Financial Disclosure’ sections do not match. 

Response 3

The discrepancy has been corrected.

Comment 4

We note that you have indicated that data from this study are available upon request. PLOS only allows data to be available upon request if there are legal or ethical restrictions on sharing data publicly. For more information on unacceptable data access restrictions, please see http://journals.plos.org/plosone/s/data-availability#loc-unacceptable-data-access-restrictions

Response 4

We have now added the following statement to the Methods section:

“The deidentified data that support the findings of this study are available on reasonable request from the corresponding author, MS. The data are not publicly available due to legislative restrictions, as the data contains information that could compromise the privacy of the research participants.”

 

Responses to comments made by the reviewer #1

Comment 1

Abstract and Statement of Significance

There are several grammatical errors in the Abstract and the Statement of Significance (e.g. words such as “a” and “the” appear to be missing resulting in some incomplete sentences and impaired readability, the phrase “solely depending” should be changed to “Solely dependent”). These section would benefit from a more thorough review by the authors. The Statement of Significance repeats the last line of the Abstract verbatim and does not explain the significance of this work (e.g. how does this work expand on what is already known about sleep difficulties in adults?).

Response 1

We have re-checked the language of Abstract and Statement of Significance and made several corrections.

Comment 2

Introduction

There are grammatical errors throughout the Introduction section (as in the Abstract, words such as “a” and “the” appear to be missing; e.g. lines 76-77). Some sentences are awkwardly worded such that readability is impacted (e.g. line 77-78)

Methods

There are grammatical errors throughout the Methods section (as in the Abstract, words such as “upon”; e.g. line 122). The Methods section would benefit from a review for errant punctuation and general grammatical/spelling errors (e.g. line 104-10).

Discussion

There are grammatical errors throughout the Discussion section (e.g. lines 166-167, line 174, line 175, line 177, lines 181-183, line 198).

Line 178 – The term “instrument” should be replaced with a more accurate term such as “technique,” as the analysis is not physical tool or device as is suggested by the term instrument.

Line 201 – change “may” to “should”

Response 2

We have re-checked the language and made several corrections.

Comment 3

Methods

There is no description of whether informed consent (or a waiver alternative) was obtained from participants.

Response 3

We have now added/modified the following text to the Methods section:

“All the respondents have approved a written informed consent. The study did not include minors. The ethics committee of the Hospital District of Helsinki and Uusimaa approved the study plan and the informed consent form (registration number HUS/1210/2016).”

Comment 4

Methods

How were physical activity and alcohol consumption dichotomous values determined? If this is based on established norms, citations should be included. Additionally, it might be more accurate to use different cut offs for excessive alcohol consumption based on participant characteristics (e.g. sex or BMI).

Response 4

As indicated in the Methods section:

“The level of physical activity was calculated from the survey responses, converted into metabolic equivalent of task (MET) and dichotomized based on the cohort’s quartiles as “low physical activity” – the lower quartile vs. others. Alcohol consumption was obtained from the survey and converted into g/week, and >210 g of pure alcohol per week was considered a cut-off for excess alcohol consumption (no/yes).”

Comment 5

Methods

There is no citation or description or established psychometric properties of the Jenkins Sleep Scale, nor is there a citation for the method for handling missing data.

Response 5

The following text has now been added to the Methods section:

“The JSS has been one of the most commonly used questionnaires in epidemiological sleep studies 1-4. It has bene found to be valid and reliable amongst patients with different health problems as well as in large non-clinical populations [25-28].”

We have now modified the following text to the Methods section adding a new reference:

“In order to include also incomplete responses to the JSS, in this study, a total score was substituted by the average score of answered items (0.0 to 5.0) using a method of person mean imputation [25].”

Heymans M, Eekhout I. Missing data in questionnaires. 2019 [cited September 22, 2021]. In: Applied missing data analysis with SPSS and (R) studio [Internet]. Amsterdam: Heymans and Eekhout, [cited September 22, 2021]. Available from: https://bookdown.org/mwheymans/bookmi/

Comment 6

Methods

After introducing the JSS abbreviation, continue to use it through out rather than switching back and forth between the abbreviation and spelling out the full name of the measure in the text.

Response 6

This has been corrected as suggested.

Comment 7

Methods

What methods or variables were used to determine the six clusters? What do the clusters consist of? How are they similar or different?

Response 7

We have now extended the text of the Methods as follows:

“The number of groups may be defined by the size of the data set measuring in two dimensions: the number of cases and the number of repeated measures. There are no common recommendations on the number of trajectory groups. In theory, it can be any number from one up to the number of cases. Previous research has suggested breaking a sample down below 300 cases may not add significant information. In this study, we pre-agreed that the smallest group should be around 3% of the entire sample or the size of the smallest group should be around 300 cases [33]. We also pre-agreed that the number of trajectory groups will be the same for all gender-age groups to ease the interpretation of the results. This way, six cluster groups were identified for each gender-age group.”

A new reference has been added:

Nagin D, Tremblay R. Developmental trajectory groups: fact or a useful statistical fiction? Criminology. 2005;43(4):873-904.

Comment 8

Results

It is unclear to me how the cluster models/trajectories (shown in Table 1) relate to the risk factor analyses (shown in Table 2). Further description of the purpose of and relationship between these analyses in either the Methods/Statistical Analysis section or in the Results would be helpful.

Response 8

We have now added the following text to the Results:

“Table 1 illustrates the path of defining the final set of six trajectory groups. The table shows that the goodness of fit increased with every step from one- to six-group model. Six-cluster models with cubic regression demonstrated good fit for each gender-age group.”

Comment 9

Tables and Figures

Table 1: the text “The chosen models are shown in bold” belongs in the footnote/caption rather than in the title.

Tables and Figures

Table 2: the text “Two trajectories with lowest baseline JSS scores were combined into one cluster (“steadily good sleepers”) and used as a reference” belongs in the footnote/caption rather than in the title. Additionally, highlighting significant findings in some manner (e.g. bolding text) would be helpful to the reader.

Response 9

The modifications have now been made as suggested.

Comment 10

Tables and Figures

The line graph Figures are completely unreadable as the text is blurry.

Response 10

Unfortunately, this is something coming from a journal submission system.

---

## [Decision Letter · Decision Letter 1]

21 Oct 2021

Persistence of sleep difficulties for over 16 years amongst 66,948 working-aged adults

PONE-D-21-02910R1

Dear Dr. Saltychev,

We’re pleased to inform you that your manuscript has been judged scientifically suitable for publication and will be formally accepted for publication once it meets all outstanding technical requirements.

**As suggeste by a reviewer, in editing, the text "% confidence limits are shown as dot-lines" in the Results section in reference to the placement of Figure 1 appears to not have been removed with the rest of a phrase that was deleted from this revised version of the manuscript. This may need to be removed/clarified. **

Kind regards,

Federica Provini

Academic Editor

PLOS ONE

Additional Editor Comments (optional):

Reviewers' comments:

Reviewer's Responses to Questions

**Comments to the Author**

1. If the authors have adequately addressed your comments raised in a previous round of review and you feel that this manuscript is now acceptable for publication, you may indicate that here to bypass the “Comments to the Author” section, enter your conflict of interest statement in the “Confidential to Editor” section, and submit your "Accept" recommendation.

Reviewer #1: All comments have been addressed

2. Is the manuscript technically sound, and do the data support the conclusions?

Reviewer #1: Yes

3. Has the statistical analysis been performed appropriately and rigorously? 

Reviewer #1: Yes

4. Have the authors made all data underlying the findings in their manuscript fully available?

Reviewer #1: No

5. Is the manuscript presented in an intelligible fashion and written in standard English?

Reviewer #1: Yes

6. Review Comments to the Author

Reviewer #1: I thank the authors for their attention to previous comments and for the opportunity to review this interesting research. In editing, the text "% confidence limits are shown as dot-lines" in the Results section in reference to the placement of Figure 1 appears to not have been removed with the rest of a phrase that was deleted from this revised version of the manuscript. This may need to be removed/clarified.

7. PLOS authors have the option to publish the peer review history of their article (what does this mean?). If published, this will include your full peer review and any attached files.

Reviewer #1: No

---

## [Editor Report · Acceptance letter]

9 Nov 2021

PONE-D-21-02910R1 

Persistence of sleep difficulties for over 16 years amongst 66,948 working-aged adults 

Dear Dr. Saltychev:

I'm pleased to inform you that your manuscript has been deemed suitable for publication in PLOS ONE. Congratulations! Your manuscript is now with our production department. 

Kind regards, 

on behalf of

Dr. Federica Provini 

Academic Editor

PLOS ONE